# Variant in the synaptonemal complex protein SYCE2 associates with pregnancy loss through effect on recombination

Valgerdur Steinthorsdottir [1] ✉, Bjarni V. Halldorsson [1,2], Hakon Jonsson [1], Gunnar Palsson [1], Asmundur Oddsson [1], David Westergaard [3,4], Gudny A. Arnadottir [1], Lilja Stefansdottir[1], Karina Banasik [3,4], M. Sean Esplin[5], Thomas Folkmann Hansen [3,6], Søren Brunak [3], Mette Nyegaard [7], Sisse Rye Ostrowski [8,9], Ole Birger Vesterager Pedersen [9,10], Christian Erikstrup [11,12], DBDS genomics consortium*, Gudmar Thorleifsson [1], Lincoln D. Nadauld[13], Asgeir Haraldsson [14,15], Thora Steingrimsdottir[14,16], Laufey Tryggvadottir [14,17], Ingileif Jonsdottir [1,14], Daniel F. Gudbjartsson [1,18], Eva R. Hoffmann [19], Patrick Sulem [1], Hilma Holm [1], Henriette Svarre Nielsen [4,9] & Kari Stefansson [1,14] ✉

Two-thirds of all human conceptions are lost, in most cases before clinical detection. The lack of detailed understanding of the causes of pregnancy losses constrains focused counseling for future pregnancies. We have previously shown that a missense variant in synaptonemal complex central element protein 2 (*SYCE2*), in a key residue for the assembly of the synaptonemal complex backbone, associates with recombination traits. Here we show that it also increases risk of pregnancy loss in a genome-wide association analysis on 114,761 women with reported pregnancy loss. We further show that the variant associates with more random placement of crossovers and lower recombination rate in longer chromosomes but higher in the shorter ones. These results support the hypothesis that some pregnancy losses are due to failures in recombination. They further demonstrate that variants with a substantial effect on the quality of recombination can be maintained in the population.

Chromosomal abnormalities, found in 60% of pregnancy losses and only 0.1% of live births, are the most common cause of pregnancy loss[1–4]. Consequently, most chromosomal abnormalities in zygotes are incompatible with life. The risk of pregnancy loss is affected by reproductive history and is greatest in the oldest age groups[5,6]. Evidence suggests that this risk follows the rate of aneuploidy, where the highest rates are observed for mothers under the age of 20, and 33 or older[7]. This is consistent with the fact that current estimates of chromosomal abnormalities in pregnancy losses are higher than reported in earlier studies, because a greater proportion of women are now conceiving at advanced maternal age[3].

It has been estimated that a large proportion of pregnancies are lost shortly after the implantation stage before being clinically recognized[1]. These very early losses that occur before an embryo has developed are assumed to be most often due to structural malformations or chromosomal aberrations, incompatible with further development and life[8]. The majority of chromosomal abnormalities are maternally transmitted[9] in contrast to de novo mutations, most of which are of paternal origin[10–12]. Genetic studies of pregnancy losses have traditionally included a small number of individuals or families[13,14]. Recessive lethal mutations and their contribution to pregnancy losses have been

**Fig. 1 | GWAS meta-analysis of pregnancy loss in 114,761 cases and 565,604 controls.** Manhattan plot illustrating the findings from a meta-analysis of pregnancy loss. *P* values (−log₁₀) from a fixed-effects inverse variance-weighted meta-analysis for each variant of association results, calculated using logistic regression for individual datasets, are plotted against their respective positions on each chromosome. Variants with *P* < 0.001 are shown. The single genome-wide significant variant identified (SYCE2:p.His89Tyr; rs189296436) is indicated. *P* values are two-sided without Bonferroni correction.

assessed in a recent large study, identifying genes in which couples carrying loss-of-function mutations had an excess of miscarriages[15]. However, the causes of both euploid and aneuploid pregnancy losses remain largely unknown and no common genetic risk factors have been reported. Four low-frequency and rare variants were recently reported to associate with sporadic and recurrent miscarriage in a genome-wide association study (GWAS) meta-analysis[16] in a study that overlaps with our current study in the use of data from the UK Biobank (UKB). However, these results remain to be validated.

In this study we sought to find variants associating with pregnancy loss in the largest dataset investigated to date and to explore their mechanism of action.

## Results

### Association analysis of pregnancy loss

We performed genome-wide association meta-analysis on 114,761 women with pregnancy loss and 565,604 female controls from Iceland, Denmark, the United Kingdom, the United States and Finland. Cases were defined based on International Classification of Diseases (ICD) codes for spontaneous abortion, missed abortion or recurrent pregnancy loss, or self-reported pregnancy loss (Supplementary Table 1). We discovered a single variant associating with pregnancy loss, rs189296436-A, $P = 6.6 \times 10^{-12}$, odds ratio (OR) = 1.22 (95% confidence interval (95% CI), 1.16–1.30; heterogeneity *P* value ($P_{het}$) = 0.14) (Figs. 1 and 2 and Supplementary Table 2). The effect of the variant was comparable when analyzed separately in pregnancy loss defined only by ICD codes ($P = 6.1 \times 10^{-8}$; OR = 1.26 (95% CI, 1.16–1.38)) and self-reported pregnancy loss ($P = 1.7 \times 10^{-6}$; OR = 1.20 (95% CI, 1.11–1.29)) (Supplementary Table 3). No other variants associated with pregnancy loss in the current study.

Furthermore, none of the four variants previously reported to associate with sporadic and multiple consecutive miscarriage[16] associated with pregnancy loss in our dataset (*P* > 0.05) (Supplementary Table 4). Conversely, our discovery variant, rs189296436, associated with sporadic pregnancy loss in the previous study ($P = 5.7 \times 10^{-7}$, OR = 1.31 (95% CI, 1.18–1.46)). We note that there is sample overlap between the two studies. A comparison of the two studies is outlined in the Supplementary Note.

The associated variant, rs189296436, is a missense variant in *SYCE2* (NM_001105578.1:c.265C>T p.His89Tyr) with a minor allele frequency (MAF) of 0.18–1.27% in the study populations (Supplementary Table 2). SYCE2 is part of the synaptonemal complex, a protein structure that mediates alignment, synapsis (pairing) and recombination of homologous chromosomes during meiosis[17]. Together with testis-expressed protein 12 (TEX12), SYCE2 forms a fibrous midline backbone of the synaptonemal complex. SYCE2:p.His89 is located in the protein core (Fig. 3) and based on X-ray crystal structures of the human proteins, it is one of the most prominent and conserved surface-exposed amino acids of the SYCE2–TEX12 complex[18]. Furthermore, introducing a glutamate mutation of amino acid His89 partially blocks structural assembly of the SYCE2–TEX12 synaptonemal complex[18], which presumably leads to less efficient synapsis.

### Effect of SYCE2:p.His89Tyr on recombination phenotypes

We have previously shown that SYCE2:p.His89Tyr associates with recombination phenotypes[19]. We observed a genome-wide significant association of the variant with three recombination phenotypes in maternal transmission, decreased telomere distance, increased GC content and increased replication timing (Table 1). The variant did not associate with any recombination phenotypes in paternal transmission[19].

To shed further light on the effect of SYCE2p.His89Tyr on recombination, we analyzed our previously presented dataset[19] in further detail. Given the high rate of aneuploidies in pregnancy losses and heterogeneity in missegregation across the chromosomes[20,21], we conducted a detailed analysis of the impact of SYCE2:p.His89Tyr on crossovers per chromosome in maternal transmissions. We reanalyzed our crossover data and constructed the same set of phenotypes as before, but with a modified measure of the distance of crossover to telomere. In this work we measure the distance from the ends of the chromosomes as defined by the GRCh38 reference[22], whereas in our earlier publication[19] the distance was measured to the first marker used in constructing the recombination map. As in our previous study we observed the strongest association ($P = 3.5 \times 10^{-108}$) with telomere distance, where each copy of the minor allele results in crossovers being on average 0.81 s.d. closer to the telomere. We then considered the crossovers occurring on each of the 22 autosomes, separately. We note that statistics computed over a single chromosome will have greater variability than statistics computed as an average over many chromosomes, leading to less power to detect true associations and effect estimates that are not directly comparable to the genome-wide estimates.

We first considered the effect of SYCE2:p.His89Tyr on telomere distance. Figure 4a shows the average distance of crossovers from telomere, measured in megabases (Mb), in carriers versus noncarriers of SYCE2:p.His89Tyr, where there was a clear deviation from the straight line of no effect, particularly in the larger chromosomes. The strongest effect ($-0.41$ s.d. or $-6$ Mb, $P = 1.6 \times 10^{-30}$) was observed on chromosome 2 (Supplementary Table 5), where carriers of SYCE2:p.His89Tyr had a larger fraction of crossovers near telomeres and a smaller fraction near the center of the chromosome (Fig. 4b). We then considered whether the association was dependent on the chromosome length. We observe a negative correlation between the effect of this variant on distance of crossovers from the telomere and the length of the chromosome, where for each 1-Mb increase in the length of the chromosome, the difference in distance from the telomere between carriers and noncarriers of SYCE2:p.His89Tyr increased by 0.0017 s.d. ($P = 5.7 \times 10^{-10}$) or 24 kilobases (Fig. 4c and Extended Data Fig. 1).

We further examined separately the effect of SYCE2:p.His89Tyr on telomere distance in individuals with only a single crossover transmitted per chromosome and in those where more than one crossover is transmitted. The effect of the variant correlated significantly with chromosome length in both sets (Extended Data Figs. 1 and 2).

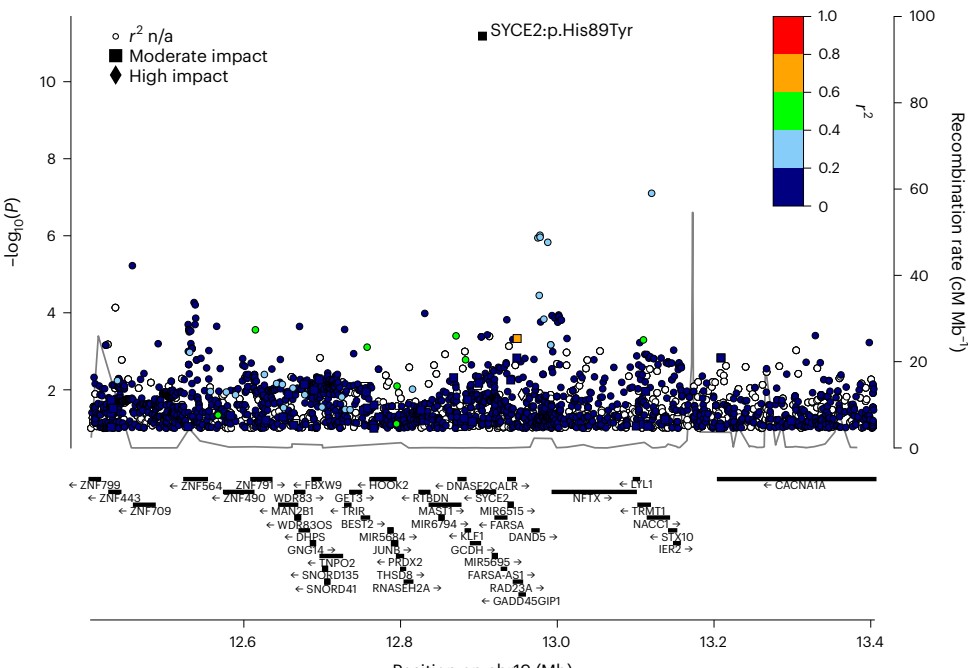

**Fig. 2 | Regional association plot for the *SYCE2* locus.** *P* values ($-\log_{10}$) of single nucleotide polymorphism associations in the pregnancy loss meta-analysis are plotted against their chromosomal positions (NCBI Build 38 coordinates). The index variant rs189296436 (SYCE2:p.His89Tyr) is indicated; other variants are colored to reflect their correlation, $r^2$, with the index variant. Known genes in the region are shown underneath the plot. *P* values are two-sided without Bonferroni correction. n/a, not available.

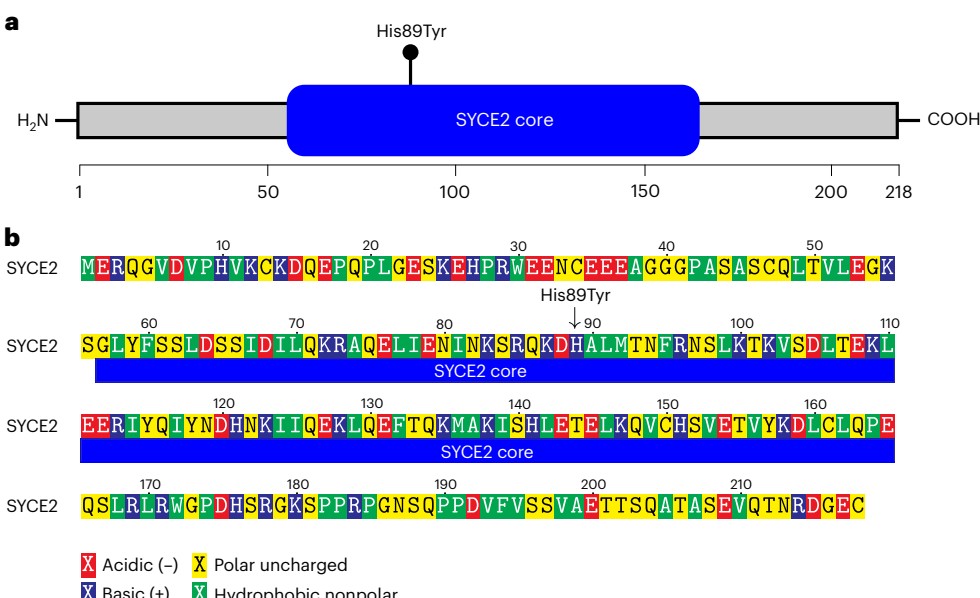

**Fig. 3 | SYCE2 protein schematic diagram. a**, The SYCE2 core (blue) (definition based on Dunce et al.[18]). The black lollipop represents the pHis89Tyr missense variant. The SYCE2 protein reference, NP_001099048.1, has 218 amino acids in total. The axis represents amino acid numbers. **b**, The black arrow indicates the pHis89Tyr missense variant we report as associating with increased risk of pregnancy loss. Functional properties of amino acids are color coded as follows: acidic residues (red), basic residues (blue), polar uncharged residues (yellow) and hydrophobic residues (green). The SYCE2 core region (blue) is indicated below the amino acid sequence.

Remarkably, although SYCE2:p.His89Tyr was only nominally associated with the genome-wide recombination rate (effect = −0.11 s.d., *P* = 0.0045) (Table 1), it associated with recombination rate on chromosome 2 at genome-wide significance (effect = −0.25 s.d. or −31 cM, *P* = 4.8 × 10$^{-12}$) (Supplementary Table 6). This pattern of association was explained by SYCE2:p.His89Tyr associating with a lower recombination rate in longer chromosomes and a higher recombination rate in smaller chromosomes (effect = −0.16 cM per 1-Mb increase in the length of chromosome, *P* = 2.6 × 10$^{-8}$) (Fig. 4d). The effects of SYCE2:p.His89Tyr did not correlate with chromosome size for GC content (*P* = 0.19), crossover hotspots (*P* = 0.89) or replication timing (*P* = 0.83) (Extended Data Fig. 3).

**Table 1 | Association of SYCE2:p.His89Tyr with recombination phenotypes in maternal transmissions**

| Phenotype | Beta (95% CI) | P value | Phenotype | Beta (95% CI) | P value |
|---|---|---|---|---|---|
| Telomere distance | −0.806 (0.735–0.877) | $3.5×10^{-108}$ | Recombination hotspots | 0.010 (0.060–0.080) | 0.78 |
| GC content | 0.365 (0.292–0.438) | $6.9×10^{-23}$ | Recombination rate | −0.107 0.093–0.110) | 0.0045 |
| Replication timing | 0.298 (0.225–371) | $1.2×10^{-15}$ | | | |

The telomere distance phenotype was recomputed for this publication (Methods) and thus the effect and *P* value differ from the published study[19].

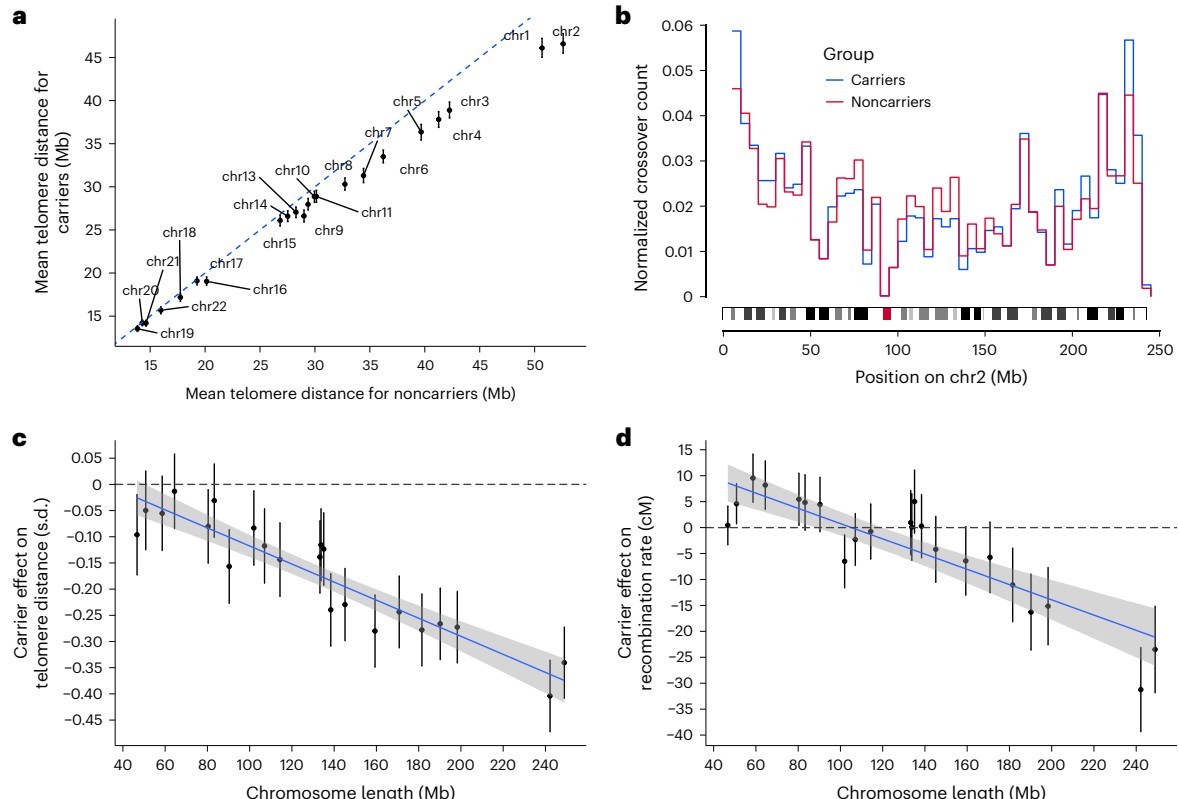

**Fig. 4 | Effect of SYCE2:p.His89Tyr on telomere distance and recombination rate. a**, Mean telomere distance for maternal crossovers on all autosomes with the data for noncarriers of SYCE2:p.His89Tyr on the *x* axis and the data for carriers on the *y* axis. The points show the mean telomere distance of all maternal crossovers transmitted to offspring. The line is *y* = *x* and the carrier effects result in a deviation from that line. Error bars (only visible for carriers) show 95% CI for the mean and are computed with bootstrapping. **b**, Distribution of crossovers on chromosome 2 for carriers (blue) and noncarriers (red) of SYCE2:p.His89Tyr. The graph shows the normalized count of crossovers within bins of size 5 Mb. **c**, Effect ($e_{TD}$) of SYCE2:p.His89Tyr on the telomere distance of crossovers, plotted against the length of the corresponding chromosome (*l*). The points indicate the mean effect of SYCE2:p.His89Tyr on the telomere distance of crossovers on each

chromosome computed with an additive association model. The error bars correspond to 95% CI of the mean from the association model. The blue line shows a linear regression fit to the model $e_{TD} \approx l$ (slope = −0.0017, *P* = 5.7 × $10^{-10}$) with 95% CI indicated by shading. **d**, The same as panel **c**, but showing the effect ($e_{RR}$) of SYCE2:p.His89Tyr on the per-chromosome recombination rate in offspring. SYCE2:p.His89Tyr does not associate significantly with the total recombination but has the effect of lowering the recombination rate on longer chromosomes and increasing it on the shorter ones. The blue line shows a linear regression fit to the model $e_{RR} \approx l$ (slope = −0.16 cM $Mb^{-1}$, *P* = 2.6 × $10^{-8}$) with 95% CI indicated by shading. Results are based on *n* = 2,932,036 autosomal crossovers observed in 70,086 maternal meioses, 1,768 where the mother is a carrier and 68,318 where the mother is a noncarrier.

The largest effect on telomere distance and recombination rate is observed on chromosome 2. However, this effect is not significantly different from the effect on chromosome 1 (Supplementary Tables 5 and 6 and Extended Data Figs. 1 and 3). Our data suggest that difference in effect on telomere distance and recombination rate is mainly driven by the size of the chromosome.

### SYCE2:p.His89Tyr and crossover interference
Crossover formation is a well-regulated process known to be under strong genetic control[23]. The formation of one crossover is known to reduce the probability of a second crossover occurring nearby under a process known as crossover interference. A subset of crossovers, however, appears to escape crossover interference during female meiosis[23].

We used the crossover data to estimate parameters of the Housworth–Stahl model[24,25]: crossover interference (*v*) and escape from crossover interference (*p*). Larger crossover interference parameter (*v*) means that the crossovers are less clustered and more evenly distributed, while *v* = 1 represents no crossover interference and random distribution of crossovers across the chromosome. High levels of the crossover escape parameter, *p*, similarly represents more random placement of crossovers across each chromosome. We estimated *v* in maternal meiosis as 6.59 and *p* as 0.039. Carriers of SYCE2:p.His89Tyr were less susceptible to crossover interference (5.97 (*v*), 0.045 (*p*)) than noncarriers (6.61 (*v*), 0.039 (*p*)) (*P* = 1.7 × $10^{-20}$). Both lower levels of crossover interference and higher levels of escape from crossover interference imply a less efficient crossover specification or maturation. A random crossover

distribution underlies aneuploidy in female meiosis[23,26] which could explain the elevated pregnancy loss in the carriers.

## Fecundity

Given the effect of SYCE2:p.His89Tyr on pregnancy loss, we wanted to determine whether it affects fecundity. We counted the number of children born to carrier and noncarrier mothers but did not find evidence that the variant affects the number of children born to 4,584 heterozygous or 18 homozygous women (Supplementary Table 7).

## Variants associating with recombination phenotypes

SYCE2:p.His89Tyr was one of 47 variants we identified that independently associate with at least one of five recombination phenotypes, when tested separately and jointly in maternally and paternally transmitted chromosomes[19]. None of the other 46 variants associated with pregnancy loss after adjusting for the number of tests ($P > 0.05/46 = 0.001$) (Supplementary Table 8). However, nominally associated markers ($P < 0.05$) were overrepresented in this group (7 of 46, $P = 0.015$, binomial test). It may be the case that only some of the phenotypes tested in the previous study are associated with reproduction. Notably, in our previously reported GWAS the strongest association for SYCE2:p.His89Tyr was observed with maternal telomere distance. The only other GWAS signal for this phenotype, C14orf39:p.Leu524Phe, also associates nominally with pregnancy loss (OR = 0.985, $P = 0.006$) (Supplementary Table 8), with an opposite effect on both telomere distance and pregnancy loss from that observed for SYCE2:p.His89Tyr, such that crossovers occurring closer to the telomere associate with a higher rate of pregnancy loss for both variants. The protein product of C14orf39, SIX6OS1, is a component of the synaptonemal complex central element and mice lacking this protein are infertile due to failure in meiosis I (ref. 27). Three homozygous loss-of-function mutations in this gene have been reported in infertile individuals[28].

We tested, with Mendelian randomization analysis[29,30], whether there is indication of a causal relationship between the traits, using as instruments variants that associate with individual recombination traits as exposure and pregnancy loss as outcome, but did not see evidence in support of this (Extended Data Fig. 4).

## Discussion

The aim of this study was to increase our understanding of factors leading to the loss of pregnancy. Synaptonemal complex proteins are key elements in meiosis and, therefore, important for reproductive success. Rare familial variants have been reported in patients with premature ovarian insufficiency or nonobstructive azoospermia where gamete production is affected, resulting in sub- and infertility[31]. However, candidate gene studies have not revealed robust evidence of association of variants in synaptonemal complex genes with pregnancy loss and/or chromosomal abnormalities[32]. In contrast, our hypothesis-free GWAS has yielded a variant in this biologically important structure that associates with pregnancy loss. The variant, SYCE2:p.His89Tyr, is located in a protein-protein interaction site that is critical for assembly of the central element of the synaptonemal complex, a key factor in mediating synapsis and recombination during meiosis.

Here we report the effect of SYCE2:p.His89Tyr on recombination and pregnancy loss. Our results support the hypothesis that a proper formation of crossovers is essential for the development of the embryo. Recombination in distal chromosomal regions has been associated with increased risk of aneuploidy[33,34], which is consistent with our findings that recombination occurs on average closer to the telomeres in carriers of SYCE2:p.His89Tyr. The variant also associates with recombination rate where the effect is correlated with chromosomal length. Recombination rate is related to the incidence of aneuploidy, where aneuploid oocytes and embryos have been shown to have lower recombination rates than euploid ones[35]. This suggests that recombination on the larger chromosomes in particular may be less stable in carriers of SYCE2:p.His89Tyr, resulting in increased aneuploidy of large chromosomes.

Our evaluation of recombination patterns in the Icelandic population requires that the transmitted crossovers result in viable offspring[19,36,37]. The association of SYCE2:p.His89Tyr with both pregnancy loss and recombination phenotypes, especially of the larger chromosomes, suggests that a fraction of crossovers from carriers of the variant result in early pregnancy loss. Pregnancy losses included in this study are based on clinical diagnosis or self-report and it seems fair to assume that most will have occurred between 6 and 20 weeks of gestation. Our data do not include early losses or those that occur around the time of implantation since these generally go unnoticed and no such datasets are available to our knowledge. We propose that the effect of SYCE2:p.His89Tyr on recombination that we have measured in live born individuals, that is, pregnancies that survive, may be more extreme in those pregnancies that are lost and may indeed contribute to the pregnancy loss. Abnormalities of the largest chromosomes provide an explanation of only a small fraction of aneuploidies detected in pregnancy losses[20]. However, they are detected at higher rates before this stage as seen in biopsies from preimplantation embryos[38], suggesting that these aneuploidies are more deleterious and may be lost at very early stages, even before a pregnancy can be detected. The effect of SYCE2:p.His89Tyr on pregnancy loss may thus be an underestimation in our study.

In summary, we have discovered an association between a missense variant in SYCE2 and pregnancy loss in a residue that is important for the assembly of the synaptonemal complex[18], an essential component of meiosis. We further show that the variant associates with crossover interference, the distance of recombination from telomeres and recombination rate on chromosomes transmitted from carrier mothers, and this effect is correlated with the length of the chromosome. We propose that this variant affects pregnancy loss through increased rate of chromosomal abnormalities. Given that the main effect of the variant on recombination is on the larger chromosomes, we speculate that, similarly, the effect on aneuploidy may also be biased towards the larger chromosomes. As a result, pregnancy loss due to SYCE2:p.His89Tyr may occur at very early stages, and the effect of this variant may, therefore, be underestimated in clinical and self-reported pregnancies. This finding offers insight into the process of meiotic recombination and the mechanisms underlying pregnancy loss.

## Online content

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

¹deCODE genetics/Amgen, Inc., Reykjavik, Iceland. ²School of Technology, Reykjavik University, Reykjavik, Iceland. ³Novo Nordisk Foundation Center for Protein Research, Faculty of Health and Medical Sciences, University of Copenhagen, Copenhagen, Denmark. ⁴Department of Obstetrics and Gynecology, Copenhagen University Hospital, Hvidovre, Copenhagen, Denmark. ⁵Division of Maternal and Fetal Medicine, Intermountain Health, Murray, UT, USA. ⁶Danish Headache Center & Danish Multiple Sclerose Center, Department of Neurology, Copenhagen University Hospital, Rigshospitalet-Glostrup, Copenhagen, Denmark. ⁷Department of Health Science and Technology, Faculty of Medicine, Aalborg University, Aalborg, Denmark. ⁸Department of Clinical Immunology, Copenhagen University Hospital, Rigshospitalet, Copenhagen, Denmark. ⁹Department of Clinical Medicine, Faculty of Health and Medical Sciences, University of Copenhagen, Copenhagen, Denmark. ¹⁰Department of Clinical Immunology,

Zealand University Hospital, Køge, Denmark. [11]Department of Clinical Immunology, Aarhus University Hospital, Aarhus, Denmark. [12]Department of Clinical Medicine, Aarhus University, Aarhus, Denmark. [13]Precision Genomics, Intermountain Health, Saint George, UT, USA. [14]Faculty of Medicine, University of Iceland, Reykjavik, Iceland. [15]Children's Hospital Iceland, Landspitali University Hospital, Reykjavik, Iceland. [16]Department of Obstetrics and Gynecology, Landspitali University Hospital, Reykjavik, Iceland. [17]Icelandic Cancer Society Research and Registration Center, Reykjavik, Iceland. [18]School of Engineering and Natural Sciences, University of Iceland, Reykjavik, Iceland. [19]Department of Cellular and Molecular Medicine, Faculty of Health and Medical Sciences, University of Copenhagen, Copenhagen, Denmark. *A list of authors and their affiliations appears at the end of the paper. ✉e-mail: valgerdur.steinthorsdottir@decode.is; kstefans@decode.is

## DBDS genomics consortium

David Westergaard[3,4], Karina Banasik[3,4], Søren Brunak[3], Ole Birger Vesterager Pedersen[9,10], Christian Erikstrup[11,12], Thomas Folkmann Hansen[3,6], Sisse Rye Ostrowski[8,9], Mette Nyegaard[7], Daniel F. Gudbjartsson[1,18] & Kari Stefansson[1,14]

A full list of members and their affiliations appears in the Supplementary Information.

## Methods

### Ethics

Our study complies with all relevant ethical regulations and was approved by relevant local authorities. The Icelandic study was approved by the Icelandic National Bioethics Committee (approval no. VSN-19-023). All participants who donated blood signed an informed consent form. The Copenhagen Hospital Biobank (CHB) Reproduction Study was approved by the National Committee on Health Research Ethics (NVK-1805807) and the Capital Region Data Protection Agency (P-2019-49). The genetic study under the Danish Blood Donor Study (DBDS) was approved by the Danish National Committee on Health Research Ethics (NVK-1700407) and the Capital Region Data Protection Agency (P-2019-99)[39]. The North West Research Ethics Committee reviewed and approved UKB's scientific protocol and operational procedures (REC reference no.: 06/MRE08/65). The Intermountain Healthcare Institutional Review Board approved the US study and all participants provided written, informed consent before enrollment. The Coordinating Ethics Committee of the Helsinki and Uusimaa Hospital District evaluated and approved the FinnGen research project. The project complies with existing legislation (in particular, the Biobank Law and the Personal Data Act). The official data controller of the present study is the University of Helsinki.

### Study populations

In the Icelandic part of the study the mean birth year of cases was 1949 (interquartile range (IQR) 1930–1965) and controls 1970 (IQR 1950–2000). Variants identified through whole-genome sequencing (WGS) of 63,460 individuals were imputed into 173,025 chip-genotyped Icelanders using long-range phasing and their untyped close relatives based on genealogy[40,41]. The personal identities of the participants and biological samples were encrypted by a third-party system.

The Danish study group consisted of participants in the CHB Reproduction Study and blood donors from the DBDS. The CHB Reproduction Study involves a targeted selection of patients with reproductive phenotypes from the CHB, a biobank based on patient blood samples drawn in Danish hospitals[42]. Mean birth year of cases was 1970 (IQR 1960–1980) and controls 1974 (IQR 1964–1986). The Danish study samples were chip typed at deCODE genetics and genotypes were imputed using a North European sequencing panel of 25,215 individuals (including 8,360 Danes).

The UKB project is a large prospective cohort study of 500,000 individuals from across the United Kingdom, aged between 40 and 69 years at recruitment[43]. Mean birth year of cases was 1952 (IQR 1945–1958) and controls 1951 (IQR 1945–1957). The study has collected extensive phenotypic and genotypic information on participants, including ICD10-coded diagnoses from hospital records, primary care data as well as detailed questionnaire data. Genotype imputation data were available for 431,079 individuals of European origin imputed with a reference panel based on WGS of around 150,000 individuals[44]. The UKB resource was used under application no. 56270. All phenotype and genotype data were collected following an informed consent being obtained from all participants.

The US study participants were recruited by the Intermountain HerediGene and Inspire studies. HerediGene is a population study aiming to recruit 500,000 participants to examine the genetic causes of diseases, in a large-scale collaboration between Intermountain Healthcare, deCODE genetics and Amgen, Inc. Inspire is Intermountain's active registry for the collection of biological samples, clinical information, laboratory data and genetic information, from consenting patients. Over 30,000 people have joined the registry. Mean birth year of cases was 1980 (IQR 1974–1987) and controls 1961 (IQR 1947–1977). Samples underwent WGS using NovaSeq Illumina technology ($n = 16,661$) and were genotyped using Illumina GSA chips ($n = 68,992$) at deCODE genetics, then filtered on 98% variant yield and duplicate samples removed. A phased haplotype reference panel was prepared from the sequence

variants using the long-range phased chip genotype data and variants identified through WGS were imputed into 61,120 chip-genotyped individuals using in-house tools and methods[41,45]. All individuals included in this study were genetically determined to be of European descent.

Finnish data originated from the FinnGen database, consisting of samples collected from the Finnish biobanks, and phenotype data collected at the national health registers. FinnGen summary statistics for data freeze 8 were imported in December 2022 from a source available to researchers (https://www.finngen.fi/en/access_results)[46].

### Phenotype definition

The pregnancy loss case group consisted of 114,761 women from Iceland, Denmark, the United Kingdom, the United States and Finland with clinical diagnosis of pregnancy loss from electronic health records or self-reported pregnancy loss (Supplementary Table 3). Clinical diagnosis included spontaneous abortion (ICD10:O03; ICD9;634; ICD8:643), recurrent pregnancy loss (ICD10:N96, O262; ICD9:6298; ICD8:6430) and missed abortion (ICD10:O021; ICD9:632; ICD8:634, 6451). Self-reported cases from Iceland completed a pregnancy history questionnaire when participating in a nation-wide cohort study of the Cancer Detection Clinic of the Icelandic Cancer Society, carried out in connection with routine population screening for cancers of the cervix and breast over a 30-year period (1964–1994). Participants were asked if they had experienced a miscarriage, and, if so, how many times. Women who reported at least one miscarriage were included in the study. Self-reported cases from the United Kingdom were women who participated in the UKB study and answered a touchscreen question 'How many spontaneous miscarriages?' (data field 3839) with 1 or more.

The control groups consisted of women from each study excluding cases.

### Association testing and meta-analysis

We used logistic regression to test for association of sequence variants with pregnancy loss in the Icelandic, Danish, US and UK datasets separately, assuming an additive genetic model, using software developed at deCODE genetics[41]. In the Icelandic analysis we included county of birth, age, age squared and an indicator function for the overlap of the lifetime of the individual with the time span of phenotype collection as covariates to account for differences between cases and controls. When analyzing the Danish, US and UK data, age and the first 20 principal components were included as covariates. We used linkage disequilibrium score regression to account for distribution inflation due to cryptic relatedness and population stratification in each of the cohorts[47].

For the meta-analyses, we combined GWASs from the respective cohorts using a fixed-effects inverse variance method based on effect estimates and standard errors, in which each dataset was assumed to have a common OR but allowed to have different population frequencies for alleles and genotypes. Sequence variants were mapped to NCBI Build 38 and matched on position and alleles to harmonize the datasets. After excluding variants with discrepant allele frequency between cohorts, as well as variants with MAF < 0.01% or imputation info < 0.8 in all cohorts, 49,932,846 variants were included in the meta-analysis.

Genome-wide significance was determined using class-based Bonferroni significance thresholds, adjusting for all variants tested[48]. Sequence variants were split into five classes based on their genome annotation, with significance threshold for each class based on the number of variants in that class (for example, lowest thresholds for high-impact variants and highest for low-impact variants). The adjusted significance thresholds are $1.31 \times 10^{-7}$ for variants with high impact (including stop-gain and loss, frameshift, splice acceptor or donor, and initiator codon variants), $2.62 \times 10^{-8}$ for missense or splice-region variants and in-frame indels, $2.38 \times 10^{-9}$ for low-impact variants (synonymous, 5′ and 3′ untranslated regions, upstream and downstream variants), $1.19 \times 10^{-9}$ for other low-impact variants in DNase

I hypersensitivity sites (intronic, intergenic, regulatory-region) and $3.97 \times 10^{-10}$ for all other variants not in DNase I hypersensitivity sites (intronic, intergenic).

## Statistics and reproducibility

No statistical method was used to predetermine sample size. Women with the relevant diagnosis or self-reported phenotype were included as cases. Women who were not included in the case group were used as controls. Males were excluded from the control groups in the GWAS association analysis. Variants with discrepant allele frequency between cohorts, as well as variants with MAF < 0.01% or imputation info < 0.8 in all cohorts, were excluded from the analysis. The experiments were not randomized. The investigators were not blinded to allocation during experiments and outcome assessment.

## Recombination phenotypes

We previously identified 4,531,535 crossovers in 126,427 meioses[19], 70,037 maternal and 56,390 paternal, with the goal of constructing a recombination map. For each meiosis the locations of all crossovers transmitted to the offspring were identified. Five different phenotypes were constructed from the crossovers transmitted from a parent to its offspring and associations were performed between the parents' genotypes and these phenotypes. Phenotypes were constructed in each sex separately and then tested in the sexes separately and jointly, for a total of 15 GWASs. The same phenotypes were used in the current study with the exception that the definition of the telomere distance was slightly modified.

With the exception of telomere distance, the phenotypes were processed as described in ref. 19. All phenotypes were rank-normal transformed before association testing. Phenotypes for each chromosome were computed in an analogous manner, considering only the crossovers that occurred on the given chromosome.

The phenotypes tested were:

Recombination rate (RR): the number of crossovers transmitted from parent to offspring. Individuals carrying markers associating with an increased recombination rate transmitted chromosomes with an increased number of crossovers.

Recombination hotspots (RH): the fraction of crossovers occurring in regions where the recombination rate is 10× the genomic average recombination rate. Individuals carrying markers associating with increased recombination hotspot rate transmit crossovers that occur more frequently in recombination hotspots.

Telomere distance (TD): the average normalized distance of crossovers from the nearest telomere. In this work we measure the distance from the ends of the chromosomes as defined by the GRCh38 reference[22], whereas in our earlier publication[19] the distance was measured to the first marker used in constructing the recombination map. Individuals carrying markers associating with increased telomere distance transmit crossovers that occur further from the telomere.

GC content (GC): the average GC content in a 1,000-base pair window near the crossovers. Individuals carrying markers associating with increased GC content transmit crossovers that occur in regions of higher GC content.

Replication timing (RT): the average replication timing score of the crossovers. Individuals carrying markers associating with increased replication timing score transmit crossovers that have an increased replication timing score, signifying that they occur in earlier replicating regions.

We refer to the four latter phenotypes as 'location phenotypes' as they are indicative of where the crossovers are located within the chromosome, but have been normalized with respect to the number of crossovers that occur within a chromosome.

## Crossover interference

Crossover interference parameters were computed using the function fitStahl in the software package xoi[49], using data described in ref. 19. The data consisted of crossovers for 70,035 maternal meioses for each of the 22 autosomes. In 1,766 meioses the mother was a carrier of SYCE2:p.His89Tyr and in 68,269 a noncarrier. We tested the null hypothesis that crossovers of carriers and noncarriers obey the same distribution in crossover interference parameters against the alternative that they were governed by two distributions, one for carriers and the second for noncarriers. We ran fitStahl separately in three groups: for all maternal meioses, for maternal meioses where the mother was a carrier of SYCE2:p.His89Tyr and for maternal meioses where the mother was a noncarrier of SYCE2:p.His89Tyr. fitStahl computes optimal values of the crossover interference parameters along with the likelihood of the observed crossover data under the Housworth–Stahl model[24]. We then computed a P value, assuming Wilk's theorem, using a likelihood ratio test, under a chi-squared distribution with two degrees of freedom.

## Interaction with chromosome length

Interaction of recombination phenotypes with chromosome length in SYCE2:p.His89Tyr carriers was computed by first associating SYCE2:p.His89Tyr with the phenotype in question using the association pipeline described in ref. 19. The effect estimates for each chromosome and their variance were used as input into a linear regression using the function lm in R[50].

## Reporting summary

Further information on research design is available in the Nature Portfolio Reporting Summary linked to this article.

## Data availability

GRCh38.p1: https://www.ncbi.nlm.nih.gov/datasets/genome/GCF_000001405.27/. FinnGen summary statistics were obtained at https://www.finngen.fi/en/access_results. The GWAS summary statistics for the pregnancy loss meta-analysis are deposited at https://www.decode.com/summarydata/. Source data are provided with this paper.

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

## Acknowledgements

We acknowledge the Novo Nordisk Foundation for grants no. NNF17OC0027594 (D.W., K.B., T.F.H. and S.B.) and no. NNF14CC0001 (D.W., K.B., T.F.H. and S.B.). The funders had no role in study design, data collection and analysis, decision to publish or preparation of the manuscript. We thank the participants and investigators of the FinnGen study.

## Author contributions

V.S., H.S.N. and K.S. conceived the study. V.S., H.H., D.W., K.B., T.F.H., S.B., M.N., S.R.O., O.B.V.P., C.E., H.S.N., L.D.N., M.S.E., I.J., L.T., A.H. and T.S. carried out data collection and subject ascertainment and recruitment. G.P. and B.V.H. analyzed recombination data. V.S., H.J., A.O., G.A.A., L.S. and G.T. analyzed pregnancy loss data. V.S., B.V.H., H.J. and K.S. wrote the paper with input from G.P., A.O., G.A.A., P.S., H.H., D.F.G., D.W., H.S.N. and E.R.H. V.S. and K.S. supervised the study. All authors approved the final version of the paper.

## Competing interests

V.S., B.V.H., H.J., G.P., A.O., G.A.A., L.S., G.T., D.F.G., H.H., P.S., I.J. and K.S. are employees of deCODE genetics, a subsidiary of Amgen.

## Additional information

**Extended data** is available for this paper at https://doi.org/10.1038/s41594-023-01209-y.

**Correspondence and requests for materials** should be addressed to Valgerdur Steinthorsdottir or Kari Stefansson.

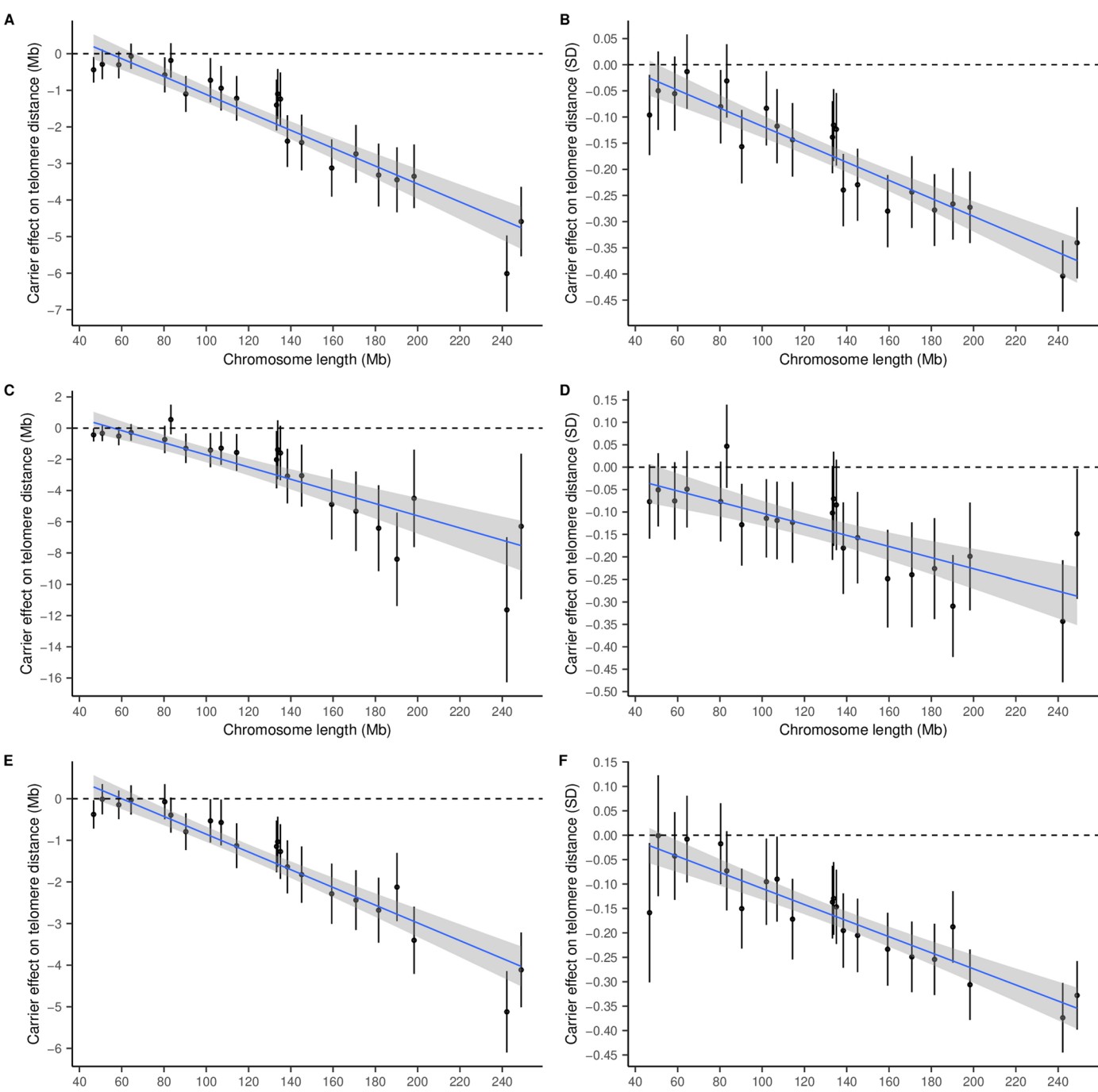

**Extended Data Fig. 1 | Effect of SYCE2:p.His89Tyr on mean telomere distance of crossovers. a**) Effect ($e_{TD}$) of SYCE2:p.His89Tyr on the mean telomere distance of crossovers, plotted against the length ($l$) of the corresponding chromosome. The points indicate the mean effect of SYCE2:p.His89Tyr on the telomere distance of crossovers on each chromosome computed with an additive association model. The error bars correspond to 95% CI of the mean from the association model. The blue line shows a linear regression fit to the model $e_{TD} \sim l$ (slope = −0.024 Mb per Mb, p-value = 3.6 × 10⁻¹¹) with 95% CI indicated by shading. **b**) Same as A but with effects rank normalized for each chromosome (slope =

−0.0017 per Mb, p-value = 5.7 × 10⁻¹⁰). **c, d**) Same as A-B using data where there is a single crossover per proband per chromosome (on left: slope = −0.0039 Mb per Mb, p-value = 8.0 × 10⁻⁸, on right: slope = −0.0012 per Mb, p-value = 2.2 × 10⁻⁵). **e, f**) Same as A-B using data where there are at least two crossovers per proband per chromosome (on left: slope = −0.0021 Mb per Mb, p-value = 1.5 × 10⁻¹¹, on right: slope = −0.0016 per Mb, p-value = 2.3 × 10⁻⁹). Results based on n = 2,932,036 autosomal crossovers observed in 70,086 maternal meioses, 1,768 where the mother is a carrier and 68,318 where the mother is a non-carrier.

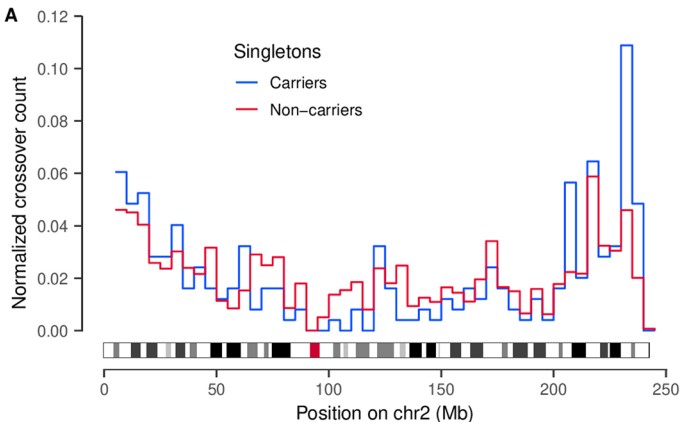

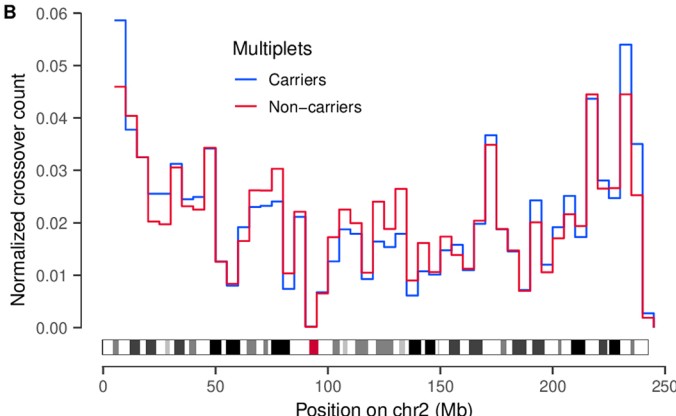

**Extended Data Fig. 2 | Distribution of crossovers. a**) Distribution of maternal crossovers on chromosome 2 for carriers (blue) and non-carriers (red) of SYCE2:p.His89Tyr. The graph shows the normalized count of crossovers within bins of size 5 Mb, where the data is restricted to probands with only a single crossover on the chromosome. **b**) Same as A but with the crossover data restricted to probands with at least two crossovers on the chromosome. Data are depicted for chromosome 2 as it shows the greatest effect of the variant on telomere distance. Data in panel A comprises 7,172 meioses, 252 where the mother is a carrier and 6,920 where the mother is a non-carrier. Data in panel B comprises 61,378 meioses, 1,449 where the mother is a carrier and 59,929 where the mother is a non-carrier.

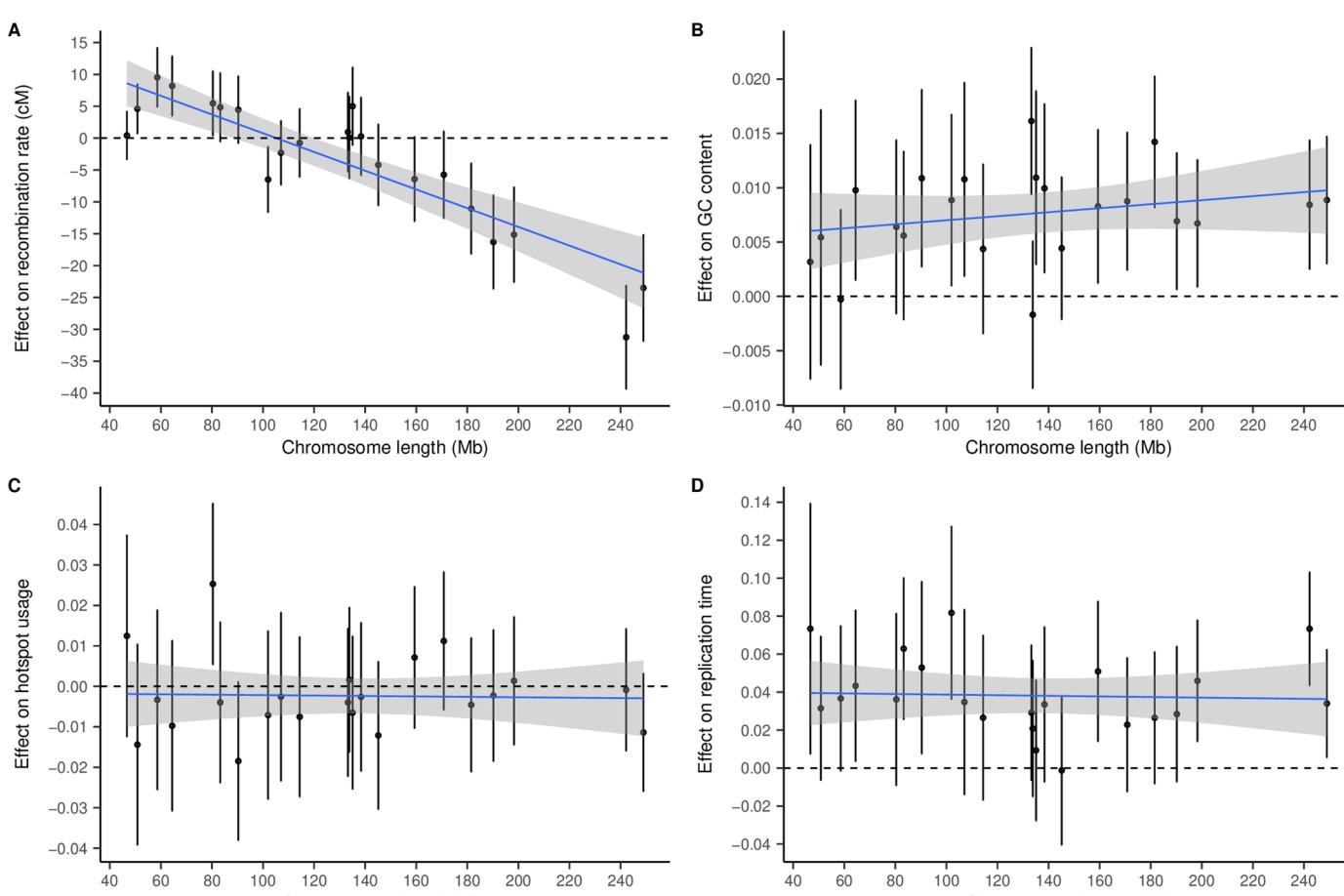

**Extended Data Fig. 3 | Carrier effect of SYCE2:p.His89Tyr on the per-chromosome characteristics of maternal crossovers in the offspring of the carrier.** The effects are plotted against the length ($l$) of chromosome, with points showing the mean effect on each chromosome and the error bars indicating 95% CI for the mean from the association models. A) Effect ($e_{RR}$) on recombination rate. The blue line shows a linear regression fit to the model $e_{RR} \sim l$ (slope = −0.0016 cM/Mb, p-value = $2.6 \times 10^{-8}$) with 95% CI indicated by shading. B) Effect ($e_{GC}$) on GC content within 500 bases of the median location of the crossover. Linear regression fit to the model $e_{GC} \sim l$ does not show significant variation with

the length of the chromosome. C) Effect ($e_{RH}$) on recombination hotspot usage (hotspots as defined in ref. 19). Linear regression fit to the model $e_{RH} \sim l$ does not show significant variation with the length of the chromosome. D) Effect ($e_{RT}$) on replication timing value (dataset GM12878 from ref. 51) at median crossover location. Linear regression fit to the model $e_{RT} \sim l$ does not show significant variation with the length of the chromosome. Results based on $n$=2,932,036 autosomal crossovers observed in 70,086 maternal meioses, 1,768 where the mother is a carrier and 68,318 where the mother is a non-carrier.

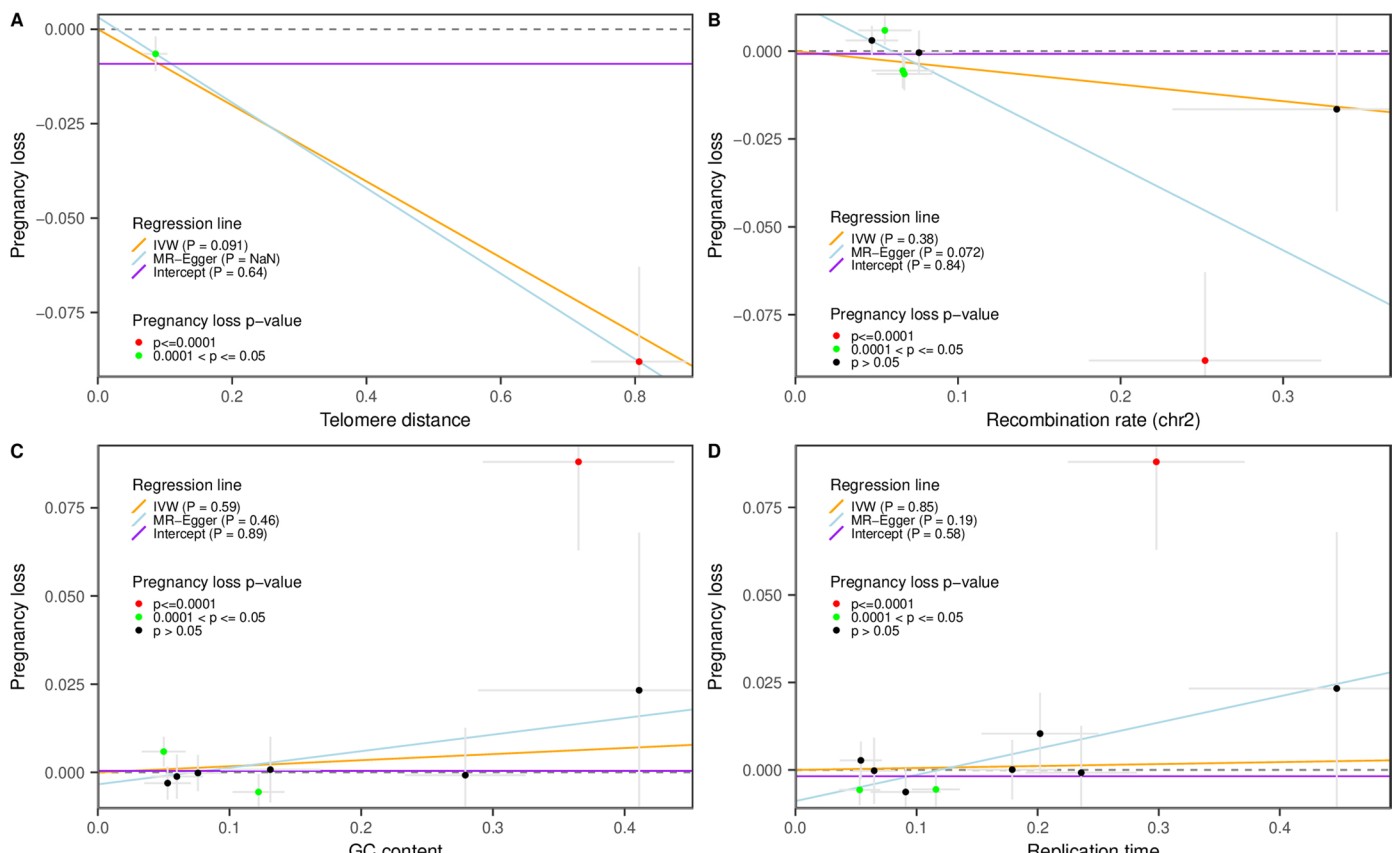

**Extended Data Fig. 4 | Mendelian randomization.** The figure shows **a**) Mendelian randomization of variants associating with telomer distance as exposure and pregnancy loss as outcome. The orange line, and the corresponding P-value, corresponds to linear regression without an intercept term, weighted by the inverse-variance of the outcome associations (inverse-variance weighted, IVW); the blue line is a weighted linear regression with an intercept term (MR-Egger); and the purple line a weighted linear regression with an intercept term only. The variants are colored according to the significance of their association with pregnancy loss in the meta-analysis and the crosses indicate 95% confidence intervals. Panels **b**, **c** and **d** show the same for B) variants associating with recombination rate as exposure, C) variants associating with GC content as exposure and D) variants associating with replication time as exposure.

# Reporting Summary

## Statistics

For all statistical analyses, confirm that the following items are present in the figure legend, table legend, main text, or Methods section.

| n/a | Confirmed | |
|---|---|---|
| ☐ | ☒ | The exact sample size (*n*) for each experimental group/condition, given as a discrete number and unit of measurement |
| ☒ | ☐ | A statement on whether measurements were taken from distinct samples or whether the same sample was measured repeatedly |
| ☐ | ☒ | The statistical test(s) used AND whether they are one- or two-sided <br> *Only common tests should be described solely by name; describe more complex techniques in the Methods section.* |
| ☐ | ☒ | A description of all covariates tested |
| ☐ | ☒ | A description of any assumptions or corrections, such as tests of normality and adjustment for multiple comparisons |
| ☐ | ☒ | A full description of the statistical parameters including central tendency (e.g. means) or other basic estimates (e.g. regression coefficient) AND variation (e.g. standard deviation) or associated estimates of uncertainty (e.g. confidence intervals) |
| ☐ | ☒ | For null hypothesis testing, the test statistic (e.g. *F*, *t*, *r*) with confidence intervals, effect sizes, degrees of freedom and *P* value noted <br> *Give P values as exact values whenever suitable.* |
| ☒ | ☐ | For Bayesian analysis, information on the choice of priors and Markov chain Monte Carlo settings |
| ☒ | ☐ | For hierarchical and complex designs, identification of the appropriate level for tests and full reporting of outcomes |
| ☒ | ☐ | Estimates of effect sizes (e.g. Cohen's *d*, Pearson's *r*), indicating how they were calculated |

*Our web collection on statistics for biologists contains articles on many of the points above.*

## Software and code

Policy information about availability of computer code

| Data collection | No software was used for data collection |
|---|---|
| Data analysis | R (version 4.2.2, lm version 4.2.2), (version 3.3.3, xoi – version 0.67-1)  (3.6.3, glmmTMB version 1.1.7), python 3.8.1 (and packages numpy 1.24.2, pandas 1.4.0, scipy 1.10.1, statsmodels 0.13.2) |

For manuscripts utilizing custom algorithms or software that are central to the research but not yet described in published literature, software must be made available to editors and reviewers. We strongly encourage code deposition in a community repository (e.g. GitHub). See the Nature Portfolio guidelines for submitting code & software for further information.

## Data

Policy information about availability of data

All manuscripts must include a data availability statement. This statement should provide the following information, where applicable:
- Accession codes, unique identifiers, or web links for publicly available datasets
- A description of any restrictions on data availability
- For clinical datasets or third party data, please ensure that the statement adheres to our policy

GRCh38.p1: https://www.ncbi.nlm.nih.gov/datasets/genome/GCF_000001405.27/. FinnGen summary statistics were obtained at https://www.finngen.fi/en/access_results. The GWAS summary statistics for the pregnancy loss meta-analysis are deposited at https://www.decode.com/summarydata/

## Research involving human participants, their data, or biological material

Policy information about studies with human participants or human data. See also policy information about sex, gender (identity/presentation), and sexual orientation and race, ethnicity and racism.

| | |
|---|---|
| Reporting on sex and gender | The GWAS meta-analysis included females with reported pregnancy loss and female population controls. Analysis of recombination traits included males and females. |
| Reporting on race, ethnicity, or other socially relevant groupings | Individuals included in the study were of European origin. |
| Population characteristics | The mean birth year of Icelandic cases was 1949 (interquartile range (IQR) 1930-1965) and controls 1970 (IQR 1950-2000). Mean birth year of Danish cases was 1970 (IQR 1960-1980) and controls 1974 (IQR 1964-1986). Mean birth year of UK cases was 1952 (IQR 1945-1958) and controls 1951 (IQR 1945-1957). Mean birth year of USA cases was 1980 (IQR 1974-1987) and controls 1961 (IQR 1947-1977). |
| Recruitment | For each contributing study, individuals with the following ICD codes, (ICD10:O03; ICD9;634; ICD8:643), (ICD10:N96, O262; ICD9:6298; ICD8:6430), or (ICD10:O021; ICD9:632; ICD8:634, 6451) or self reported pregnancy loss were included as cases. For each study, the control group consisted of females not included in the case group. |
| Ethics oversight | The Icelandic study was approved by the Icelandic National Bioethics Committee. The CHB Study was approved by the National Committee on Health Research Ethics and the Capital Region Data Protection Agency. The DBDS study was approved by the Danish National Committee on Health Research Ethics and the Capital Region Data Protection Agency. The North West Research Ethics Committee approved UK Biobank study. The Intermountain Healthcare Institutional Review Board approved the USA study. The Coordinating Ethics Committee of the Helsinki and Uusimaa Hospital District approved the FinnGen research project. |

Note that full information on the approval of the study protocol must also be provided in the manuscript.

# Field-specific reporting

Please select the one below that is the best fit for your research. If you are not sure, read the appropriate sections before making your selection.

☒ Life sciences    ☐ Behavioural & social sciences    ☐ Ecological, evolutionary & environmental sciences

For a reference copy of the document with all sections, see nature.com/documents/nr-reporting-summary-flat.pdf

# Life sciences study design

All studies must disclose on these points even when the disclosure is negative.

| | |
|---|---|
| Sample size | For the GWAS meta-analysis we combined the largest samples size of pregnancy loss phenotypes available within the contributing studies. |
| Data exclusions | For each contributing study samples and variants were excluded based on well established sample and variant quality control procedures to remove poor quality samples and variants.<br>For the GWAS meta-analysis variants with minor allele frequency < 0.01% or imputation info < 0.8 in all cohorts, as well as variants with discrepant allele frequency between cohorts, were further excluded. |
| Replication | The GWAS meta-analysis included all study material for the relevant phenotypes available to us so we did not conduct replication. There was no evidence for heterogeneity between studies indicating that the results were not driven by a single false positive study. |
| Randomization | The case group comprised all women who fulfilled the diagnostic criteria and the control group comprise all women who did not fulfill the diagnostic criteria. No randomization was applied. |
| Blinding | This is an observational study and no blinding was required |

# Reporting for specific materials, systems and methods

We require information from authors about some types of materials, experimental systems and methods used in many studies. Here, indicate whether each material, system or method listed is relevant to your study. If you are not sure if a list item applies to your research, read the appropriate section before selecting a response.

## Materials & experimental systems

| n/a | Involved in the study |
|---|---|
| ☒ | Antibodies |
| ☒ | Eukaryotic cell lines |
| ☒ | Palaeontology and archaeology |
| ☒ | Animals and other organisms |
| ☒ | Clinical data |
| ☒ | Dual use research of concern |
| ☒ | Plants |

## Methods

| n/a | Involved in the study |
|---|---|
| ☒ | ChIP-seq |
| ☒ | Flow cytometry |
| ☒ | MRI-based neuroimaging |

