## [Peer Review File · Nature Structural & Molecular Biology]

Peer Review Information

Manuscript Title: Variant in the synaptonemal complex protein SYCE2 associates with pregnancy loss through effect on recombination

Corresponding author name(s): Valgerdur Steinhorsdottir, Kari Stefansson

Reviewer Comments & Decisions:

Decision Letter, initial version:

Message: 11th Jul 2023

Dear Dr. Steinhorsdottir,

Please accept my sincere apologies in the very long delay in communicating a decision on your study. I am afraid that, in addition to the difficulty in obtaining suitable referee reports, I have been on medical leave for the past weeks.

Thank you again for submitting your manuscript "Variant in the synaptonemal complex protein SYCE2 associates with pregnancy loss through effect on recombination". We now have comments (below) from the 3 reviewers who evaluated your paper. In light of those reports, we remain interested in your study and would like to see your response to the comments of the referees, in the form of a revised manuscript.

You will see that all three referees appreciate the study and the timely conclusions. Nevertheless, they have some comments and suggestions to improve the study. Specifically, please revise the study to expand and strengthen analysis as suggested by Reviewer #1, improve on statistical analysis and compare to previous studies as suggested by Reviewers #2 and #3, and provide additional detail on the study population. Please be sure to address and respond to all concerns of the referees in full in a point-by-point response and highlight all changes in the revised manuscript text file. If you have comments that are intended for editors only, please include those in a separate cover letter.

We expect to see your revised manuscript within 6 weeks. If you cannot send it within this time, please contact us to discuss an extension; we would still consider your revision, provided that no similar work has been accepted for publication at NSMB or published elsewhere.

Reporting Summary:

Please note that all key data shown in the main figures as cropped gels or blots should be presented in uncropped form, with molecular weight markers. These data can be aggregated into a single supplementary figure item. While these data can be displayed in a relatively informal style, they must refer back to the relevant figures. These data should be submitted with the final revision, as source data, prior to acceptance, but you may want to start putting it together at this point.

Data availability: this journal strongly supports public availability of data. All data used in accepted papers should be available via a public data repository, or alternatively, as Supplementary Information. If data can only be shared on request, please explain why in your Data Availability Statement, and also in the correspondence with your editor. Please note that for some data types, deposition in a public repository is mandatory - more information on our data deposition policies and available repositories can be found below: <https://www.nature.com/nature-research/editorial-policies/reporting-standards#availability-of-data>

[Redacted]

Sincerely,

Carolina

Carolina Perdigoto, PhD

Chief Editor
Nature Structural & Molecular Biology
orcid.org/0000-0002-5783-7106

Reviewers' Comments:

Reviewer #1:

Remarks to the Author:

The manuscript by Steinthorsdottir et al describes an association between a missense variant in SYCE2 and pregnancy loss. To my knowledge this is the largest genetic study to date for this important and under-studied trait. Overall I thought the paper was well written and the analyses are well conducted. I have a few comments and questions for the authors:

1. It wasn't immediately clear to me what the difference in phenotype (and sample overlap) was between what is studied here and the miscarriage work by Laisk et al. I note the signals in Laisk et al do not replicate here – does this SYCE2 signal replicate in the other study? Given the relative paucity of findings here a meta-analysis of the two studies (or some other form of data integration) would clearly be desirable.
2. The authors may wish to leverage eQTL / pQTL resources etc to identify other putatively associated variants at sub-threshold associations. There may well be interesting observations to highlight that could be leveraged with such data.
3. Is it possible to explore at what stages in pregnancy the SYCE2 variant is associated with? Is it only very early pregnancy loss or associated with later stages also?
4. It would be interesting for the authors to perform Mendelian Randomization analyses on the various recombination / telomere traits against pregnancy loss, rather than just testing the variants individually. This could naturally be extended to other modifiable risk factors such as BMI, smoking, alcohol etc.

Reviewer #2:

Remarks to the Author:

This is an interesting study investigating maternal genetic factors that may influence risk of pregnancy loss in a large sample of patients (114,761 individuals reporting pregnancy loss; 565,604 controls). The authors identify a single rare variant exceeding the threshold for statistical significance. The variant is a mis-sense mutation in the gene SYCE2, which is a component of the synaptonemal complex, involved in chromosome pairing and recombination during meiosis. Interestingly, the variant had previously appeared in an earlier study of recombination by the deCODE Genetics group as associated with various features of the recombination landscape. The authors thus hypothesize that the mechanism by which the variant drives pregnancy loss is by impacting recombination and predisposing oocytes to aneuploidy. While this is not directly demonstrated, the story is very compelling, and it is rare to have even this level of mechanistic detail for a GWAS.

The manuscript is interesting, novel, and well-organized. The results are of immediate

interest to the fields of reproductive genetics and human development, as well as anyone studying the proteins and molecular biology underlying processes such as meiosis and cell division. The comments and suggestions below are relatively minor, primarily regarding the organization of the paper, as well as an additional statistical analysis; these should not prohibit its publication.

Major comments/questions:

1. Is the ploidy status of the pregnancy losses used as phenotypes in the primary association study known? One hypothesis raised by this study is that the association should be driven by aneuploid pregnancy losses in particular, however this information may not be available.
2. Related to the point above, the introduction of the manuscript discusses the high rates of pregnancy loss shortly after implantation, before pregnancies are clinically recognized. However, to be considered as a phenotype in the current association study, presumably the pregnancy would have to be clinically recognized. Thus, these represent later pregnancy losses. Can the authors comment on how these observations can be reconciled?
3. In the Results section, I recommend restructuring the initial paragraph (lines 99-104) to more clearly frame (1) what question is being assessed in this section, (2) what data is being used, and (3) how this differs from the previous publication.
4. Please describe the genotyping methods in greater detail. Given that the variant is rare and not in LD with other nearby variants, quality of genotyping is crucial. We therefore recommend reporting quality control metrics of the genotyping at that particular site.
5. More information about the frequency of the allele of interest in relevant populations could strengthen the major conclusion. What is the allele frequency of this variant in different populations of published datasets (e.g., 1000 Genomes Project)? Is the variant restricted to particular populations? What is the frequency of this variant in the datasets used here? What is the frequency of this variant in each population represented by these datasets?

Minor comments:

1. The caveat and claims in the third paragraph (lines 196-207) are reasonable and well-supported. Recommend a few changes to make this text more clear and valuable:
 - Move the last sentence (lines 205-207) earlier in the paragraph
 - "Indicates" (line 198) is too strong; recommend switching to "suggests", or adding more evidence or context to support the claim that it "indicates"
 - Rephrase sentence in lines 201-203 to clarify that while aneuploidies affecting larger chromosomes are depleted in detected pregnancy losses, they are detected at higher rates prior to this stage (e.g., biopsies from pre-implantation embryos), suggesting that these aneuploidies are less survivable/more deleterious
2. Include an age range for the young and advanced age mothers (line 60)
3. Expand upon the findings from that past study and connect them to this current work (lines 70-71)

4. Add context for what that past GWAS found and how the current study differs (lines 73-74)
5. Add citation for “which is consistent with crossovers occurring closer to the telomere being associated with a higher rate of pregnancy loss” (lines 171-172)
6. Expand upon the mechanism through which this variant is essential for mouse fertility (line 173)
7. Restructure the first two sentences of the Discussion to better frame the question being investigated here (lines 178-179).
8. In the discussion, offer an explanation for why chromosome 2 has the largest effect on recombination phenotypes
9. Extended Data Figure 4 & 5: Mention how many chromosomes were included in the first panel series (1+ CO per proband) and how many were included in the second (2+ CO per proband)
10. Extended Data Figure 5: Recommend explaining in the legend why they chose to depict chromosome 2
11. In the Methods, consider whether the ages of the mothers distributed similarly across the case and control groups in each of the datasets used.

Reviewer #3:

Remarks to the Author:

This manuscript presents the results of the largest GWAS of pregnancy loss performed to date encompassing 114,761 cases and 555,604 controls. The analysis pinpoints significant association to one single rare missense variant in SYCE2, a gene encoding for a component of the synaptonemal complex. The study is well performed and well written with interesting follow-up analyses of recombination phenotypes associated with the missense variant. I have a few comments and concerns.

The authors refer to previously published GWAS-results of sporadic and recurrent miscarriage by Laisk et al. Nat Commun 11, 5980 (2020), and further present data to show that none of the 4 loci reported by Laisk are associated with pregnancy loss in the current study. This apparent lack of replication gives rise to some concern, and I suggest that the authors address this discrepancy in more detail and deliberate on the potential explanations underlying the divergent results.

1. Both studies appear to utilize overlapping data from UK Biobank and the pregnancy loss phenotypes in the two studies appear to be similar. Thus, it would be helpful if the authors presented a summary on the overlap of datasets and differences in case definitions between the current study and the study by Laisk et al.
2. Laisk et al reported genome-wide significant association at rs146350366 with sporadic miscarriage in European ancestry cases, of which more than 37,000 were from UK

Biobank. In stark contrast, the current study shows no association with pregnancy loss at this locus (OR 0.96, $p=0.12$). I would suggest that the authors provide data on the effect estimates in the individual cohorts included in their replication analyses presented Supplementary Table 4 to illustrate how the individual cohort or cohorts contribute to the discrepancy.

3. It would be reassuring to see independent support for the lead pregnancy loss association reported in the current study. Such association data could be obtained from the non-overlapping Northern European cohorts included in the study by Laisk. These data would presumably be made available upon request.

4. The authors suggest that their association finding may facilitate genetic counselling of mothers carrying the SYCE:p.His89Tyr variant. Given that this is a yet not validated association finding with an OR of only 1.2, the statement seems too strong. I would suggest a more moderate interpretation.

Author Rebuttal to Initial comments

RESPONSE TO REVIEWERS' COMMENTS

We thank the reviewers for constructive comments.

Throughout, reviewers' comments are highlighted in bold. Changes to the text have been underlined and pointers to the specific sections of the main text are indicated.

Reviewer #1:

Remarks to the Author:

The manuscript by Steinhorsdottir et al describes an association between a missense variant in SYCE2 and pregnancy loss. To my knowledge this is the largest genetic study to date for this important and under-studied trait. Overall I thought the paper was well written and the analyses are well conducted. I have a few comments and questions for the authors:

1. It wasn't immediately clear to me what the difference in phenotype (and sample overlap) was between what is studied here and the miscarriage work by Laisk et al. I note the signals in Laisk et al do not replicate here – does this SYCE2 signal replicate in the other study? Given the relative paucity of findings here a meta-analysis of the two studies (or some other form of data integration) would clearly be desirable.

Laisk et al analysed sporadic miscarriage in European samples including 49,996 cases and 174,109 female controls. The overlap between our study and the Laisk study likely extends to the 37,105 cases from UK Biobank included in their study, as well as the corresponding controls. Given the sample overlap it was not feasible to meta-analyse the two studies and the approximately 12,000 non-

overlapping cases would not be a substantial addition to the current study of 114,761 cases. We do, however, note that in the Laisk study the SYCE2 signal associates with sporadic miscarriage with $P = 5.7 \times 10^{-7}$, OR = 1.31, consistent with our results. We have updated the paragraph on p. 3 and added a Supplementary Note, comparing the two studies. The updated text (line 89) now reads:

Furthermore, none of the four variants previously reported to associate with sporadic and multiple consecutive miscarriage¹⁸ associated with pregnancy loss in our dataset ($P > 0.05$; Supplementary Table 4). Conversely, our discovery variant, rs189296436, associated with sporadic pregnancy loss in their study ($P = 5.7 \times 10^{-7}$, OR = 1.31, 95% CI 1.18-1.46). We note that there is sample overlap between the two studies. A comparison of the two studies is outlined in the Supplementary Note.

2. The authors may wish to leverage eQTL / pQTL resources etc to identify other putatively associated variants at sub-threshold associations. There may well be interesting observations to highlight that could be leveraged with such data.

Given that SYCE2 is primarily a part of a meiotic protein complex and the effect we are reporting is based on events that take place in female meiosis, occurring early in the development of the female fetus, we do not have access to the relevant tissue to measure gene expression. SYCE2 is expressed in testis but no eQTLs are reported in testis in GTex data. The protein is not targeted by Olink or SomaLogic proteomics platforms so we are not aware of any pQTL data.

3. Is it possible to explore at what stages in pregnancy the SYCE2 variant is associated with? Is it only very early pregnancy loss or associated with later stages also?

For the majority of cases included in our study (around 90%) we have no information on the timing of the pregnancy loss. Given the low frequency of the variant and modest OR, a large number of cases are needed to explore how it relates to the timing of pregnancy loss. We assume that the majority of the pregnancy loss cases in our study occur between 6 and 20 weeks, unfortunately, we have no information on pregnancy loss in very early pregnancy where we postulate the variant may have the strongest effect. See also response to comment 2 from reviewer 2.

4. It would be interesting for the authors to perform Mendelian Randomization analyses on the various recombination / telomere traits against pregnancy loss, rather than just testing the variants individually. This could naturally be extended to other modifiable risk factors such as BMI, smoking, alcohol etc.

We agree with the reviewer that it is of interest to explore if there is a causal relationship between the recombination traits and pregnancy loss. The SYCE2 variants showed a genome-wide significant association with four recombination phenotypes when transmitted from the mother, recombination rate on chr2, telomere distance, GC content near crossovers and replication timing score of crossovers. We tried to do a Mendelian Randomization analysis using as instruments variants that

associate with the recombination traits and pregnancy loss as outcome. Unfortunately, there are only few variants associating with each of the recombination traits and we did not get meaningful results. In particular, only two variants, SYCE2 and C14orf39:p.Leu524Phe, also mentioned in the manuscript, associate with telomere distance. We further explored genetic correlation between those recombination traits and pregnancy loss using LD-score regression but got no significant results. It is unclear if we lack power to detect a relationship between those traits or if a simple relationship does not exist.

Reviewer #2:**Remarks to the Author:**

This is an interesting study investigating maternal genetic factors that may influence risk of pregnancy loss in a large sample of patients (114,761 individuals reporting pregnancy loss; 565,604 controls). The authors identify a single rare variant exceeding the threshold for statistical significance. The variant is a mis-sense mutation in the gene SYCE2, which is a component of the synaptonemal complex, involved in chromosome pairing and recombination during meiosis. Interestingly, the variant had previously appeared in an earlier study of recombination by the deCODE Genetics group as associated with various features of the recombination landscape. The authors thus hypothesize that the mechanism by which the variant drives pregnancy loss is by impacting recombination and predisposing oocytes to aneuploidy. While this is not directly demonstrated, the story is very compelling, and it is rare to have even this level of mechanistic detail for a GWAS.

The manuscript is interesting, novel, and well-organized. The results are of immediate interest to the fields of reproductive genetics and human development, as well as anyone studying the proteins and molecular biology underlying processes such as meiosis and cell division. The comments and suggestions below are relatively minor, primarily regarding the organization of the paper, as well as an additional statistical analysis; these should not prohibit its publication.

Major comments/questions:

1. Is the ploidy status of the pregnancy losses used as phenotypes in the primary association study known? One hypothesis raised by this study is that the association should be driven by aneuploid pregnancy losses in particular, however this information may not be available.

The reviewer raises an important point here. Unfortunately, we do not have any information regarding the ploidy of the pregnancy losses included in the study.

2. Related to the point above, the introduction of the manuscript discusses the high rates of pregnancy loss shortly after implantation, before pregnancies are clinically recognized. However, to be considered as a phenotype in the current association study, presumably the pregnancy would have to be clinically recognized. Thus, these represent later pregnancy losses. Can the authors comment on how these observations can be reconciled?

The cases included in this study are either based on clinical diagnosis or self-report. While we have little information on the timing of the pregnancy loss it seems fair to assume that most of the losses will have occurred between 6 and 20 weeks of gestation. Some earlier losses may be included, in particular based on self-report, but these data do not include losses that occur around the time of implantation. The reason for that is simply that these early losses generally go unnoticed so no such datasets are available to our knowledge. What we are measuring in this study is thus the effect of the SYCE2 variant on pregnancy loss that occurs largely after 6 weeks of gestation. When studying pregnancy loss, it is important to consider what is known about pregnancy loss from conception to birth, even if datasets suitable for large scale genetic studies are only available for the latter part of that period. The effect of the SYCE2 variant on recombination, in particular of the larger chromosomes, is measured in live born individuals, i.e., pregnancies that survive. We propose that these effects may be more extreme in those pregnancies that are lost and may indeed contribute to the pregnancy loss. We have no direct evidence that the SYCE2 variant causes aneuploidy but based on the effect on recombination we propose that aneuploidy is a plausible mechanism by which the variant increases the risk of pregnancy loss. Aneuploidies of larger chromosomes are not well tolerated and have been reported to be preferentially lost very early in pregnancy. What we have direct evidence for is that the SYCE2 variant associates with pregnancy loss (at 6 to 20 weeks gestation) through effect on recombination. We speculate that the effect on earlier pregnancy loss may be higher, through increased rate of aneuploidy of larger chromosomes. Unfortunately, we have no way of testing this hypothesis at this stage. One way to test this might be by studying preimplantation embryos from carrier mothers. However, this is not an easy task, and the low frequency of the variant is an added complication.

3. In the Results section, I recommend restructuring the initial paragraph (lines 99-104) to more clearly frame (1) what question is being assessed in this section, (2) what data is being used, and (3) how this differs from the previous publication.

To address this point we have added the following sentence to this paragraph (line 110):

To shed further light on the effect of SYCE2p.His89Tyr on recombination we analyzed our previously presented dataset⁴ in further detail.

We also added the following sentence from the methods (line 115):

In this work we measure the distance from the ends of the chromosomes as defined by the GRCh38 reference⁴⁶, whereas in our earlier publication⁴ the distance was measured to the first marker used in constructing the recombination map.

The same dataset was used as in our previous publication.

4. Please describe the genotyping methods in greater detail. Given that the variant is rare and not in LD with other nearby variants, quality of genotyping is crucial. We therefore recommend reporting quality control metrics of the genotyping at that particular site.

As described in the methods section the association analysis of this variant is based on imputed rather than directly genotyped data. The imputation is, however, based on large population specific WGS reference sets which allows us to reliably impute rare variants. These reference sets include 25,215 for Danes, 16,661 for the US cohort, 63,460 Icelanders, 149,697 of the UK Biobank cohort and 8,554 in FinnGen. With the exception of FinnGen the whole genome sequencing used for the imputation was performed at deCODE. The marker was not flagged for quality in any of the cohorts. Markers are flagged for quality when 1) They are in a region with > 2x average depth 2) Hardy-Weinberg equilibrium p-value < 1e-7 3) imputation information < 0.8 4) marker is reported as low quality in sequence variant calling 5) marker is in a repeat sequence 5) Sequence info < 0.6 or Sequence info > 1.4 6) Imputation yield < 70% 7) Genotyping yield < 80%. Imputation info for individual cohorts (ranging from 0.96-1.00) is reported in Supplementary Table 2. No heterogeneity was observed between cohorts for this variant.

5. More information about the frequency of the allele of interest in relevant populations could strengthen the major conclusion. What is the allele frequency of this variant in different populations of published datasets (e.g., 1000 Genomes Project)? Is the variant restricted to particular populations? What is the frequency of this variant in the datasets used here? What is the frequency of this variant in each population represented by these datasets?

The frequency of the SYCE2 variant in each dataset used here is presented in Supplementary Table 2. The highest frequency is observed in Iceland (MAF 1.27%) and the lowest in Finland (MAF 0.18%), while the frequency in the UK, USA and Danish data ranges from 0.51-0.72%. This is consistent with gnomAD data where the MAF is reported to be 0.55% in North-western Europeans and 0.13% in the Finnish population. The variant is not observed in East Asians but reported in other gnomAD population groups with the highest frequency observed in non-Finnish Europeans (0.54%).

Minor comments:

1. The caveat and claims in the third paragraph (lines 196-207) are reasonable and well-supported.

Recommend a few changes to make this text more clear and valuable:

- Move the last sentence (lines 205-207) earlier in the paragraph
- “Indicates” (line 198) is too strong; recommend switching to “suggests”, or adding more evidence or context to support the claim that it “indicates”
- Rephrase sentence in lines 201-203 to clarify that while aneuploidies affecting larger chromosomes are depleted in detected pregnancy losses, they are detected at higher rates prior to this stage (e.g.,

biopsies from pre-implantation embryos), suggesting that these aneuploidies are less survivable/more deleterious

We have edited the text in this paragraph for clarity as suggested by the reviewer

2. Include an age range for the young and advanced age mothers (line 60)

We have added maternal age range to this sentence (line 60):

Evidence suggests that this risk follows the rate of aneuploidy, where the highest rates are observed for mothers under the age of 20, and 33 or older⁹

3. Expand upon the findings from that past study and connect them to this current work (lines 70-71)

We have added more detail on this study. The relevance is simply to show that in spite of various efforts to study aspects of pregnancy loss our understanding is still very limited. The sentence now reads (line 70):

Recessive lethal mutations and their contribution to pregnancy losses have been assessed in a recent large study, identifying genes in which couples carrying loss of function mutations had an excess of miscarriages¹⁷.

4. Add context for what that past GWAS found and how the current study differs (lines 73-74)

We have updated this sentence, it now reads (line 74): Four low frequency and rare variants were recently reported to associate with sporadic and recurrent miscarriage in a GWAS meta-analysis¹⁸ in a study that overlaps with our current study in the use of data from the UK Biobank.

See also response to remark 1 from reviewer 1 for further details.

5. Add citation for “which is consistent with crossovers occurring closer to the telomere being associated with a higher rate of pregnancy loss” (lines 171-172)

We have now updated this sentence as it was clearly misleading. The sentence now reads (line 184) „such that crossovers occurring closer to the telomere associate with a higher rate of pregnancy loss for both variants“.

6. Expand upon the mechanism through which this variant is essential for mouse fertility (line 173)

We have expanded this sentence as suggested (line 185):

The protein product of C14orf39, SIX6OS1, is a component of the synaptonemal complex central element and mice lacking this protein are infertile due to failure in meiosis ¹²⁶.

7. Restructure the first two sentences of the Discussion to better frame the question being investigated here (lines 178-179).

We have edited the Discussion as suggested.

8. In the discussion, offer an explanation for why chromosome 2 has the largest effect on recombination phenotypes

We agree that it is worth mentioning that the largest effect is observed for chromosome 2. We have added a paragraph to the results section to address this comment (line 148):

The largest effect on telomere distance and recombination rate is observed on chromosome 2. However, this effect is not significantly different from the effect on chromosome 1 (Supplementary Tables 4 and 5; Extended Data Figs. 4 and 6). Our data suggest that difference in effect on telomere distance and recombination rate is mainly driven by the size the chromosome.

9. Extended Data Figure 4 & 5: Mention how many chromosomes were included in the first panel series (1+ CO per proband) and how many were included in the second (2+ CO per proband)

We have added the number of meiosis, divided into carrier and non-carrier mothers, included in the analysis to the figure legends. For Extended Data Figure 5 we have also included the number of meiosis included in the analyses of single crossovers and at least two crossovers, separately. Note that this information cannot be reported for Extended Data Figure 4 in the same way as these numbers vary between chromosomes.

10. Extended Data Figure 5: Recommend explaining in the legend why they chose to depict chromosome 2

We have added a sentence to the figure legend to clarify the selection of chromosome 2 for presentation.

Data are depicted for chromosome 2 as it shows the greatest effect of the variant on telomere distance.

11. In the Methods, consider whether the ages of the mothers distributed similarly across the case and control groups in each of the datasets used.

We note that for the Icelandic and US datasets there is substantial age difference between cases and controls. This may affect the association results and we felt it was important to include this information clearly in the methods section. It would be ideal to have better matching between cases and controls.

However, datasets including information on pregnancy loss are not widely available and we felt it was important to be more inclusive to increase the sample size rather than discarding datasets with less-than-optimal matching between cases and controls. Importantly, we note that there is no significant heterogeneity between datasets for the reported association, indicating that this does not have a strong effect on the association results.

Reviewer #3:

Remarks to the Author:

This manuscript presents the results of the largest GWAS of pregnancy loss performed to date encompassing 114,761 cases and 555,604 controls. The analysis pinpoints significant association to one single rare missense variant in SYCE2, a gene encoding for a component of the synaptonemal complex. The study is well performed and well written with interesting follow-up analyses of recombination phenotypes associated with the missense variant. I have a few comments and concerns.

The authors refer to previously published GWAS-results of sporadic and recurrent miscarriage by Laisk et al. Nat Commun 11, 5980 (2020), and further present data to show that none of the 4 loci reported by Laisk are associated with pregnancy loss in the current study. This apparent lack of replication gives rise to some concern, and I suggest that the authors address this discrepancy in more detail and deliberate on the potential explanations underlying the divergent results.

1. Both studies appear to utilize overlapping data from UK Biobank and the pregnancy loss phenotypes in the two studies appear to be similar. Thus, it would be helpful if the authors presented a summary on the overlap of datasets and differences in case definitions between the current study and the study by Laisk et al.

We have added a Supplementary Note comparing the two studies (see response to reviewer 1).

2. Laisk et al reported genome-wide significant association at rs146350366 with sporadic miscarriage in European ancestry cases, of which more than 37,000 were from UK Biobank. In stark contrast, the current study shows no association with pregnancy loss at this locus (OR 0.96, $p=0.12$). I would suggest that the authors provide data on the effect estimates in the individual cohorts included in their replication analyses presented Supplementary Table 4 to illustrate how the individual cohort or cohorts contribute to the discrepancy.

We have updated Supplementary Table 4 to include results for individual cohorts. For rs146350366 we see nominal association in the UK Biobank data, $P=0.0047$; OR = 1.19, for the major (risk) allele G. This is lower than the OR of approximately 1.4 observed for this variant in the UK Biobank sporadic miscarriage data. We see no association for our other datasets. We have no explanation why this association does

not replicate in our data but there is no indication that the discrepancy can be traced to individual cohorts.

3. It would be reassuring to see independent support for the lead pregnancy loss association reported in the current study. Such association data could be obtained from the non-overlapping Northern European cohorts included in the study by Laisk. These data would presumably be made available upon request.

As we have now described in a Supplementary Note (see response to reviewer 1) there is sample overlap between the Laisk study and ours such that out of 49,996 sporadic miscarriage cases in the Laisk study up to 37,105 cases (from UKB) may overlap with our study. That means that only around a quarter of the cases in Laisk et al. are not included in our study. We compared the association results for rs189296436-A in the UKB part of our analysis, 48,954 cases and 183,598 controls, with the public results of Laisk et al., 49,996 cases (31,105 from UKB) and 174,109 controls. The two tests have similar power (and much sample overlap) and give similar results, $P = 5.5 \times 10^{-7}$, $OR = 1.27$ (Supplementary Table 2) vs $P = 5.7 \times 10^{-7}$, $OR = 1.31$ in Laisk et.al. This suggests that the additional samples in the Laisk study, other than those from the UKB, contribute to the association of this variant with pregnancy loss. We note though that these additional samples on their own have not much power to replicate the association with rs189296436-A. Given the number of cases and controls of European ancestry in Fig. 1 of the Laisk et al. paper, excluding UKB, and assuming $OR = 1.22$ and frequency of 0.5%, we estimate that the non-overlapping samples have around 50% power to detect association of the *SYCE2* variant. We have now added the association results for the *SYCE2* variant in the Laisk study to the manuscript.

4. The authors suggest that their association finding may facilitate genetic counselling of mothers carrying the SYCE:p.His89Tyr variant. Given that this is a yet not validated association finding with an OR of only 1.2, the statement seems too strong. I would suggest a more moderate interpretation.

We agree with the reviewer and have now deleted this paragraph.

Decision Letter, first revision:

Message: Our ref: NSMB-A47672A

27th Oct 2023

Dear Dr. Steinhorsdottir,

Thank you for submitting your revised manuscript "Variant in the synaptonemal complex protein SYCE2 associates with pregnancy loss through effect on recombination" (NSMB-A47672A). It has now been seen by the original referees and their comments are below.

The reviewers find that the paper has improved in revision, and therefore we'll be happy in principle to publish it in Nature Structural & Molecular Biology, pending minor revisions to satisfy the referees' final requests and to comply with our editorial and formatting guidelines.

Sincerely,

Carolina Perdigoto, PhD
Chief Editor
Nature Structural & Molecular Biology
orcid.org/0000-0002-5783-7106

Reviewer #1 (Remarks to the Author):

I am satisfied with the responses and have no further comments.

Reviewer #2 (Remarks to the Author):

I have carefully reviewed the revised manuscript titled "Variant in the synaptonemal complex protein SYCE2 associates with pregnancy loss through effect on recombination." The authors have adequately addressed the various concerns I previously raised, which has significantly improved the quality and clarity of the manuscript, and I endorse its publication.

A couple of minor points/suggestions for the final version:

1. Regarding the response to Reviewer 1, Question 2: Doesn't GTEX have ovary data? Is this not more relevant?
2. I recommend adding the negative results in the response to Reviewer 1, Question 4 to the Results section of the manuscript for transparency and completeness.
3. I recommend adding an abbreviated version of response to Reviewer 2, Question 2 to the Discussion section.
4. I recommend adding an abbreviated version of response to Reviewer 2, Questions 3 and 4 to the Results / Supplementary Materials.

Author Rebuttal, first revision:**RESPONSE TO REVIEWERS' COMMENTS**

We thank the reviewers for constructive comments.

Reviewers' comments are highlighted in bold. Changes to the text have been underlined and pointers to the specific sections of the main text are indicated.

Reviewer #2:**Remarks to the Author:**

I have carefully reviewed the revised manuscript titled "Variant in the synaptonemal complex protein SYCE2 associates with pregnancy loss through effect on recombination." The authors have adequately addressed the various concerns I previously raised, which has significantly improved the quality and clarity of the manuscript, and I endorse its publication.

A couple of minor points/suggestions for the final version:**1. Regarding the response to Reviewer 1, Question 2: Doesn't GTEx have ovary data? Is this not more relevant?**

Ovarian expression of SYCE2 in GTEx is low and no eQTLs reported. The reason we mention testis is that unlike females, recombination takes place in adult males. The highest SYCE2 expression by far in GTEx data is observed in testis. We note that while we have no evidence that SYCE2 affects recombination in males, we also have no evidence to the contrary.

2. I recommend adding the negative results in the response to Reviewer 1, Question 4 to the Results section of the manuscript for transparency and completeness.

In line with the reviewers request we now report the negative results of the MR analysis in the results section with the following text:

We tested, with Mendelian randomization analysis^{29,30}, whether there is indication of causal relationship between the traits, using as instruments variants that associate with individual recombination traits as exposure and pregnancy loss as outcome, but did not see evidence supporting that (Extended Data Fig. 4).

We have further added Extended Data Fig. 4 with the results from the MR analysis.

3. I recommend adding an abbreviated version of response to Reviewer 2, Question 2 to the Discussion section.

We thank the reviewer for the suggestion. We have now expanded the discussion and added the following text:

Pregnancy losses included in this study are based on clinical diagnosis or self-report and it seems fair to assume that most will have occurred between 6 and 20 weeks of gestation. Our data do not include early losses or those that occur around the time of implantation since these generally go unnoticed and no such datasets are available to our knowledge. We propose that the effect of SYCE2:p.His89Tyr on recombination that we have measured in live born individuals, i.e., pregnancies that survive, may be more extreme in those pregnancies that are lost and may indeed contribute to the pregnancy loss.

4. I recommend adding an abbreviated version of response to Reviewer 2, Questions 3 and 4 to the Results / Supplementary Materials.

Our response to reviewer 2 question 3 was to add the requested text to the results section so that has already been done and no further action is taken.

Regarding the response to question 4 we have now added the following text as a Supplementary Note:

Quality control of the SYCE2:p.His89Tyr variant rs189296436

The association analysis of rs189296436 is based on imputed rather than directly genotyped data. The imputation is based on large population specific WGS reference sets (see Methods) which allows us to reliably impute rare variants. With the exception of FinnGen the whole genome sequencing used for the imputation was performed at deCODE. The marker was not flagged for quality in any of the cohorts. Markers are flagged for quality when 1) They are in a region with > 2x average depth 2) Hardy-Weinberg equilibrium p-value < 1e-7 3) imputation information < 0.8 4) marker is reported as low quality in sequence variant calling 5) marker is in a repeat sequence 5) Sequence info < 0.6 or Sequence info > 1.4 6) Imputation yield < 70% 7) Genotyping yield < 80%. Imputation info for individual cohorts (ranging from 0.96-1.00) is reported in Supplementary Table 2. No heterogeneity was observed between cohorts for this variant.

Final Decision Letter:

Message 22nd Dec 2023

:
Dear Dr. Steinthorsdottir,

We are now happy to accept your revised paper "Variant in the synaptonemal complex protein SYCE2 associates with pregnancy loss through effect on recombination" for publication as a Article in Nature Structural & Molecular Biology.

As soon as your article is published, you can generate your shareable link by entering the DOI of your article here: `http://authors.springernature.com/share`. Corresponding authors will also receive an automated email with the shareable link

Your paper will be published online soon after we receive proof corrections and will appear in print in the next available issue. You can find out your date of online publication by contacting the production team shortly after sending your proof corrections.

Please note that *Nature Structural & Molecular Biology* is a Transformative Journal (TJ). Authors may publish their research with us through the traditional subscription access route or make their paper immediately open access through payment of an article-processing charge (APC). Authors will not be required to make a final decision about access to their article until it has been accepted. [Find out more about Transformative Journals](https://www.springernature.com/gp/open-research/transformative-journals)

Authors may need to take specific actions to achieve [compliance with funder and institutional open access mandates](https://www.springernature.com/gp/open-research/funding/policy-compliance-faqs). If your research is supported by a funder that requires immediate open access (e.g. according to [Plan S principles](https://www.springernature.com/gp/open-research/plan-s-compliance)) then you should select the gold OA route, and we will direct you to the compliant route where possible. For authors selecting the subscription publication route, the journal's standard licensing terms will need to be accepted, including [self-archiving policies](https://www.springernature.com/gp/open-research/policies/journal-policies). Those licensing terms will supersede any other terms that the author or any third party may assert apply to any version of the manuscript.

If you have any questions about our publishing options, costs, Open Access requirements,

or our legal forms, please contact ASJournals@springernature.com

Sincerely,

Carolina Perdigoto, PhD
Chief Editor
Nature Structural & Molecular Biology
orcid.org/0000-0002-5783-7106